# Safe and Effective Delivery of mRNA Using Modified PEI-Based Lipopolymers

**DOI:** 10.3390/pharmaceutics15020410

**Published:** 2023-01-26

**Authors:** Huijing Wang, Xin Liu, Xuefeng Ai, K. C. Remant-Bahadur, Teo A. Dick, Bingqian Yan, Tingting Lu, Xingliang Zhou, Runjiao Luo, Minglu Liu, Xiangying Wang, Kaixiang Li, Wei Wang, Hasan Uludag, Wei Fu

**Affiliations:** 1Institute of Pediatric Translational Medicine, Shanghai Children’s Medical Center, School of Medicine, Shanghai Jiao Tong University, 1678 Dong Fang Road, Shanghai 200127, China; 2Department of Dental Materials, Shanghai Biomaterials Research & Testing Center, Ninth People’s Hospital, College of Stomatology, Shanghai Jiao Tong University School of Medicine, Shanghai Key Laboratory of Stomatology & Shanghai Research Institute of Stomatology, National Clinical Research Center of Stomatology, No. 427, Jumen Rd., Shanghai 200011, China; 3Department of Pediatric Cardiothoracic Surgery, Shanghai Children’s Medical Center, School of Medicine, Shanghai Jiao Tong University, Shanghai 200127, China; 4Department of Chemical & Materials Engineering, Faculty of Engineering, University of Alberta, Edmonton, AB T6G 2R3, Canada; 5Department of Biomedical Engineering, Faculty of Medicine & Dentistry, University of Alberta, Edmonton, AB T6G 2R3, Canada; 6Faculty of Pharmacy & Pharmaceutical Sciences, University of Alberta, Edmonton, AB T6G 2R3, Canada; 7Shanghai Key Laboratory of Tissue Engineering, Shanghai 9th People’s Hospital, School of Medicine, Shanghai Jiao Tong University, Shanghai 200011, China

**Keywords:** modRNA, lipopolymers, biocompatibility, lysosomal escape, effective transfection

## Abstract

Chemically modified mRNA (modRNA) has proven to be a versatile tool for the treatment of various cancers and infectious diseases due to recent technological advancements. However, a safe and effective delivery system to overcome the complex extracellular and intracellular barriers is required in order to achieve higher therapeutic efficacy and broaden clinical applications. Here, we explored All-Fect and Leu-Fect C as novel transfection reagents derived from lipopolymers, which demonstrated excellent biocompatibility, efficient delivery capabilities, and a robust ability to escape the lysosomes. These properties directly increase mRNA stability by preventing mRNA degradation by nucleases and simultaneously promote efficient gene translation in vitro and in vivo. The modRNA delivered with lipopolymer vectors sustained effective transfection in mouse hearts following direct intramyocardial injection, as well as in major organs (liver and spleen) after systemic administration. No observable immune reactions or systemic toxicity were detected following the systemic administration of lipopolymer-mRNA complexes to additional solid organs. This study identified commercial reagents for the effective delivery of modRNA and may help facilitate the advancement of gene-based interventions involving the safe and effective delivery of nucleic acid drug substances.

## 1. Introduction

Messenger RNA (mRNA) vaccines have rapidly emerged as a novel platform for prophylactic use and in the treatment of certain cancers [1,2,3,4,5,6]. Despite the remarkable progress in mRNA-based therapeutics, the instability and immunogenicity of mRNA hamper its clinical applications. Recent improvements in mRNA structure and function, together with enhanced manufacturing and purification processes, have highlighted mRNA as a novel therapeutic agent [7,8,9]. Chemically modified mRNAs (modRNAs) reduce levels of immune activation by avoiding toll-like receptor recognition, display a prolonged half-life, have improved stability, and are efficiently translated [10,11,12,13,14,15,16]. Hence, modRNA has become a promising therapeutic for the treatment of a wide range of diseases arising from defective proteins, which has numerous advantages over DNA-based expression modalities [14,17,18,19,20,21,22]. Unlike plasmid DNA, modRNA can quickly and efficiently express the target protein in the cytoplasm without the need to enter the nucleus, and with minimal risk of insertional mutagenesis [1,23,24]. One key challenge still facing modRNA therapies is the safe and effective delivery of the molecules to specific target cells and tissues. The most significant barrier for naked modRNA is its vulnerability to ribonuclease enzymes and its highly anionic nature, which prevents transport across cell membranes into the cells [25]. An appropriate modRNA delivery vehicle is urgently needed to these ends, but unfortunately, efficient modRNA delivery systems are still in their infancy [16,26,27].

In recent years, viral and non-viral vectors have been explored to ferry various expression systems to the target cells [5,28,29,30,31,32]. In spite of a large number of available viral vectors, unwanted genomic integration, immunogenicity (especially at repeat doses), high production costs, potential risk of secondary carcinogenesis and limited packaging capacity have hampered their wide utilization for long-term therapeutics [32,33]. On the contrary, non-viral vectors attracted significant attention due to their excellent biocompatibility and safety, ability to undergo endocytosis at the cell membrane, and efficient encapsulation ability. At present, different types of non-viral vectors have been developed to protect the nucleic acid from nucleases and facilitate uptake into cells to translate functional proteins, such as lipid nanoparticles (LNPs), polymers, dendrimers, and others [4,19,34,35,36,37,38,39,40,41,42,43,44,45,46]. However, some commercially available lipid-based transfection reagents that display high transfection efficiency also induce toxic responses in vivo [20]. Accordingly, more suitable vectors that can deliver modRNA safely with high efficiency are urgently being sought. 

Polyethyleneimine (PEI) has emerged as the most widely studied cationic polymer as a non-viral delivery system owing to its spectrum of functional amines, the facile chemistry for further functionalization, its cost-effectiveness and safety profiles [40,42,45,47,48]. The cationic amine groups of PEI can complex with mRNA via electrostatic interactions and be packaged into virus-like (~100 nm) particles to ultimately be delivered to target cells efficiently [42]. PEI offers an excellent capacity to protect the nucleic acid against degradation, but it can also enhance cellular uptake via interactions with anionic cell surface proteoglycans, increasing the half-life of the cargo in the cytoplasm. Unfortunately, unmodified PEI is effective when molecular weight (MW) is ~25 kDa, and significant concerns have been raised for intolerable toxicities. Alternatively, small MW PEI with a similar chemical backbone to larger MW PEI has been functionalized with hydrophobic groups that led to superior delivery efficiency with acceptable biosafety [42]. These lipopolymers derived from low MW PEI were recently commercialized for a variety of uses with different nucleic acids [49,50].

The aim of this study was to identify and develop appropriate modRNA vectors that exhibit excellent biocompatibility, efficient cellular uptake and have the ability to easily escape the endosomes. A series of in vitro and in vivo investigations was conducted to screen and identify top-performing commercial polymeric vectors from modified PEIs. Our results demonstrated that the chosen vectors not only protected the modRNA from degradation and facilitated cellular uptake, but also promoted release from endosomal compartments to the cytoplasm for rapid protein expression. The toxicity of the modRNA formulations with polymeric vectors was minimal in animal models and they were well tolerated when delivered via either local or systemic administration. These results show polymeric mRNA complexes as a feasible, efficient, stable, and safe biomolecule delivery system with potential clinical applications.

## 2. Materials and Methods

### 2.1. Materials

Lipopolymer transfection reagents used in this experiment were purchased from RJH Bioscience (Edmonton, AB, Canada). Lipofectamine^TM^ MessengerMAX transfection reagent (MAX) was purchased from ThermoFisher Scientific (Waltman, MA, USA). Cell culture-related reagents such as 0.25% Trypsin-EDTA (1×), Dulbecco’s Phosphate Buffered Saline (DPBS), Dulbecco’s Modified Eagles Medium (DMEM), Penicillin-Streptomycin antibiotic, and Fetal Bovine Serum (FBS) were obtained from Life Technologies (Carlsbad, CA, USA). Opti-MEM reduced serum medium (1×) was purchased from Gibco (New York, NY, USA). The cell counting kit-8 (CCK8) was purchased from Dojindo (Kumamoto, Japan). EdU Cell Proliferation Kit with Alexa Fluor 555 was purchased from Beyotime (Shanghai, China). Plasmid DNA used in this project was purchased from Sanggon Biotech (Shanghai, China). DEPC-Treated nuclease-free water was purchased from Ambion (Austin, TX, USA). MEGAscript T7 Transcription kit and transcription cleanup kit was purchased from Ambion (USA). Lyso-Tracker Probes were purchased from Life Technologies (USA). siRNA tracker Cy3 kit purchased from Mirus (Marietta, GA, USA). The primary antibodies used in this study were CD3 (ab1669, Abcam, Cambridge, UK) and CD68 (ab955, Abcam). A Donkey Anti-Mouse IgG H&L (ab150106, Abcam) and a Donkey Anti-Rabbit IgG H&L (ab15073, Abcam) were used as secondary antibodies. Cell lines of HeLa (Serial TCHu187), 3T3 (Serial GNM6), and MCF-7 (Serial TCHu74) used in the projects were purchased from the Cell Bank of the Chinese Academy of Sciences (Shanghai, China). The human induced pluripotent stem cells (hiPSCs) were donated by Yanxin Li’s laboratory at Shanghai Children’s Medical Center’s Institute of Pediatric Translational Research.

### 2.2. modRNA Synthesis

All modRNA used in this study was produced using in vitro transcription techniques, as previously described [20,26,51]. Briefly, we constructed the plasmid DNA containing the gene of interest and transformed it into E. coli. After amplification and purification of the plasmid, it was linearized by restriction endonuclease and then amplified by PCR to generate a template for in vitro transcription. Afterward, we used the MEGAscript T7 Transcription kit to produce chemically modified mRNA (modRNA) in vitro. We purified the modRNA with a transcription cleanup kit and treated the transcripts with Antarctic phosphatase for 30 min at 37 °C to remove residual 5′-phosphates. Lastly, the modRNA was repurified and all modRNAs were quantified by Nanodrop and were stored frozen at −20 °C.

### 2.3. Lipopolymer Particle Characterization

#### 2.3.1. Measurement of Particle Size

The efficiency of gene transfection and cytotoxicity are significantly influenced by the polyplexes’ particle size and zeta potential. To determine the mean diameter size and the polydispersity index (PDI) of each sample, modRNA and different carriers were mixed at a 1:2 ratio, diluted in 200 μL distilled water, and loaded into a quartz cuvette. The mean diameter size of the polyplexes was measured at an angle of 173° at room temperature (25 °C) by dynamic light scattering using a Zetasizer Nano series (Malvern Instruments, Malvern, UK). Each experiment was repeated three times and the results were averaged.

#### 2.3.2. Zeta Potential

The zeta potential of polyplexes was measured using the Zetasizer Nano series (Malvern Instruments, UK), as previously described [4,40,46]. Before testing, it was diluted to 1 mL with deionized water according to the optimal binding ratio of vector to modRNA. Each sample is measured at least three times to take the average value to analyze the experimental results.

#### 2.3.3. Circular Dichroism Analysis

To prove the structure of the modRNA wrapped by the polymeric vectors remain unchanged, the conformation of naked modRNA and the modRNA-polymer complexes were tested by a MOS-500 Circular Dichromatic (CD, Bio-Logic Science Instruments, Grenoble, France) spectrometer in the range of 220~320 nm.

#### 2.3.4. Agarose Gel Electrophoresis

To assess the nuclease resistance capability of modRNA encapsulated in the vector, agarose electrophoresis was used to evaluate the integrity of modRNA as previously reported [40]. Either free modRNA or vector-encapsulated modRNA was added to Opti-MEM medium with 10% FBS and then incubated at 37 °C for 24 h. For 1% agarose gel electrophoresis assays, the complexes were quantified in a volume of 10 μL, and each sample was loaded with 2 μL of 6 × loading buffer, and then loaded on a 1% agarose gel. Electrophoresis was conducted for 30 min at 120 V in fresh 1 × TAE (Tris-acetate-EDTA) buffer. The gel was imaged and analyzed using a gel image system (Tanon 3500, Shanghai, China) under ultraviolet light.

#### 2.3.5. Transmission Electron Microscopy (TEM)

To test the morphology and size of the polyplexes, we used a 200 kV JEOL 2100 transmission electron microscope for analysis as previously reported [4,40,46]. First, immediately after the glow discharge treatment in a vacuum, the samples were prepared on a grid coated with carbon, a drop of the polymorph suspension was added and blotted dry with filter paper after 5 min. Then, the samples were negatively stained with a 2% uranyl acetate solution for 30 s. Particle size analysis and distribution were performed using the Fiji/ImageJ software version 2.3.0. Particle size measurement was carried out on the longest dimension of the polyplexes from images taken from different regions of the sample.

### 2.4. In Vitro Cell Assays

#### 2.4.1. Cell Cultures

HeLa cells and 3T3 cells were cultured in DMEM cell medium supplemented with 10% FBS and 1% penicillin-streptomycin antibiotic at 37 °C under a 5% CO_2_ atmosphere. The medium was replaced every 2 days and cell passaging was executed when the monolayer of adherent cells reached ~90% confluence.

#### 2.4.2. modRNA Transfection

For in vitro transfection, 2 × 10^5^ cells were seeded in a 6-well plate and cultured at 37 °C for 24 h. Then, modRNA (2 μg) and vectors (typically 2, 4, and 6 μL) were separately diluted in serum-free Opti-MEM medium and incubated for 5 min. To generate modRNA-vector complexes, the separate mixes were gently pooled together and left to incubate for 15 min at room temperature (RT). Finally, the transfection complex was added directly to the cells in a complete culture medium and incubated for 24 h for testing. To transfect modRNA in vivo, 20 μg RNA and 40 μL transfection reagent were diluted separately in 200 μL Opti-MEM medium, incubated for 5 min at room temperature, then mixed and incubated for another 20 min at RT before tail vein injection [26].

#### 2.4.3. Flow Cytometric Analysis

To quantify the extent of modRNA expression, the transfection efficiency and fluorescence intensity of modRNA were measured by flow cytometry 24 h after transfection, as previously described [26]. Cells were washed with DPBS 3 times and then treated with 0.25% trypsin-EDTA for 3 min before adding fresh medium. Cells were centrifuged at 1000 rpm for 4 min to remove trypsin and resuspended in a DPBS medium for flow cytometry analysis.

#### 2.4.4. EdU Assay

To test the cell proliferation, Click-iT EdU Imaging Kits (Beyotime, China) was used according to the manufacturer’s instructions, as previously studied [52]. Briefly, HeLa cells were planted at a density of 5 × 10^4^ cells in a 24-well plate 24 h before transfection. After transfection for 24 h, EdU was added to the medium according to the reagent instructions to make a final concentration of 10 μM. Then, cells were fixed with 4% paraformaldehyde for 15 min at room temperature and permeabilized with 0.5% Triton X-100 for 15 min. To detect EdU, the configured click reaction mixes were added to each well, incubated for 30 min at room temperature, and protected from light. Lastly, to detect the proportion of cell proliferation, nuclei were stained using Hoechst 33,342 and imaged with a Leica inverted fluorescence microscope (Leica, DMI3000B, Wetzlar, Germany).

#### 2.4.5. Cellular Uptake and Endosomal Escape Analysis

In order to observe the cellular uptake and escape from the lysosome of modRNA complexes, GFP modRNA and luciferase modRNA were labeled with the fluorophore Cy3, as in previously published studies [4,40]. Initially, cells were seeded into confocal dishes for 24 h incubation and then transfected with Cy3-conjugated modRNA. The following day, the escape of modRNA from lysosome was tracked using Lyso-Tracker™ Green DND-26 of lysosomal staining dye, in accordance with the manufacturer’s directions, and the nucleus was stained with Hoechst 33342 for 15 min at room temperature. Finally, the cells were washed three times with DPBS. The intracellular localization of modRNA complexes was observed at different times by confocal laser microscopy (Leica, TSC SP8).

#### 2.4.6. In Vitro Cytotoxicity Assay

The in vitro cytotoxicity of the vectors was assessed by the Cell Counting Kit-8 [4]. Briefly, HeLa cells were seeded into a 96-well plate at a density of 1 × 10^4^ cells per well and cultured overnight. Then, the cells were treated with GFP modRNA complexed with MessengerMAX, ALL-Fect, or Leu-Fect C at 37 °C, and 5% CO_2_ for different times, respectively. For control groups, cells were either exposed to naked modRNA or left untreated. The vector-to-modRNA mixing ratio was 3:1 and all groups were performed in five parallel wells. After different time treatments (1, 2, 3, 4, and 5 days), cells were washed with DPBS and incubated with 10 μL of CCK-8 and 90 μL of DMEM cell medium at 37 °C for 2 h. Two hours later, the absorbance of the plates was measured at wavelengths of 450 nm by a multifunctional microplate reader (BIOTEK Synergy2, BioTek Instruments, Winooski, VT, USA).

### 2.5. In Vivo Animal Experiments

#### 2.5.1. In Vivo Cytotoxicity Assessment and Histological Evaluation

In vivo biocompatibility of the vectors was assessed by histological evaluation and hematological examination, as previously described [4,26,40,46]. The study was divided into two parts: local organ injection and tail vein injection. For local injection, 50 μL vector and 50 μL serum-free Opti-MEM medium were mixed evenly and injected into pancreatic organs of 6–8 weeks old C57BL/6 male mice (purchased from Shanghai JSJ Company, Shanghai, China), and the organs were harvested 3 days later. Then, the extracted organs were embedded in paraffin, sectioned, and stained with hematoxylin and eosin (H&E). Tissue inflammation was further analyzed by immunofluorescence, which involved incubating sections with primary antibodies against CD3 and CD68 overnight at 4 °C followed by staining with the Alexa Fluor-488/555 conjugated secondary antibodies for 2 h at room temperature. Lastly, DAPI was incubated for 3 min at RT before observing and imaging the sections with the Leica DM6000 B system.

For the systemic toxicity study, C57BL/6 male mice were administered 500 μL mixed solution of reagents (50 μL) and DPBS (450 μL) by tail vein injection, once every two days for 7 injections. The mice’s body weights were also measured. On day 14, mice were sacrificed, blood was drawn, and the serum was isolated. The activities of alanine aminotransferase (ALT), aspartate aminotransferase (AST), and counts of white blood cells (WBC), lymphocytes (Lymph cell), monocyte (Mon cell), and neutrophils were measured using individual assay kits. Meanwhile, organs (including the heart, liver, spleen, lung, kidney, and intestine) were gathered for histological analysis. The organs were fixed in 4% paraformaldehyde overnight, embedded in paraffin followed by sectioning (5 μm), and stained with hematoxylin and eosin. The slides were pathologically evaluated using a Leica DM6000 B system.

#### 2.5.2. In Vivo modRNA Delivery and Expression

To study in vivo transfection efficiency of modRNA, the complexes were delivered via direct intramyocardial injection or systemic tail vein infusion [4,40,43]. Rosa26mTmG mice were anesthetized with isoflurane, intubated, and the thorax was carefully opened to locate and find the heart, as well as to place retractors into the incision to ensure clarity of view. Then, 15 μg of Cre mRNA was injected into the heart before closing the chest. After three days, hearts were harvested, frozen sections were prepared, confocal fluorescence imaging was performed, and modRNA expression was statistically analyzed using Image J software.

Additionally, the biodistribution of mRNA was also evaluated using tail vein injection. The injection was prepared by adding 20 μg of Cre modRNA and 40 µL of vector to 200 µL of the reaction mixture. Then, the solution was injected into the Rosa26mTmG mice at three different sites on their tails. Frozen sections of the liver, spleen, heart, and kidney were taken for staining to survey the effect of modRNA expression after three days.

#### 2.5.3. Frozen Sections and Immunohistochemistry

The isolated organs were washed clean of residual blood with sterile DPBS and fixed in 4% paraformaldehyde for 24 h at 4 °C. The following day, the organs were washed three times with DPBS and soaked in a 30% sucrose solution to dehydrate until they precipitated to the bottom. Then, they were embedded in OCT compound and rapidly frozen at −80 °C. Tissue sections were performed using a cryostat (CM1950 Leica, Germany) with a section thickness of 5 μm and stored at −80 °C. Before staining, sections were defrosted and dried at room temperature for 30 min, then washed three times in DPBS, and permeabilized with 0.5% Triton-X 100 solution for 30 min. Then, the slides were rinsed three times every 5 min with DPBS and blocked with 5% normal goat serum for 2 h at room temperature. The sections were then probed with the appropriate concentration of primary antibody at 4 °C overnight. Following three washes, the tissue sections were incubated for two hours at room temperature and shielded from light with the appropriate fluorescently labeled secondary antibodies. Finally, imaging was carried out using a Leica TSC SP8 laser confocal microscopy system.

### 2.6. Imaging Processing

To count the number of cells, the pictures were first processed with the Image J software version 1.8.0, as previously described [52]. The image was converted to 8-bit format prior to “hole filling” and “watershed” operations being performed. Cells were counted using the “analyze particles” command. The acquired data was analyzed and graphed using Graphpad Prism version 5.0 and Origin version 8.0.

### 2.7. Statistical Analysis

The results are presented as the mean ± standard deviation (SD). Statistical analyses were performed by *t*-test, values of ** *p* < 0.01 and *** *p* < 0.001 were regarded as statistically significant.

## 3. Results and Discussion

Modified mRNA holds significant promise for gene and recombinant protein therapies but selecting the appropriate RNA delivery vehicle remains a challenge for unlocking the potential of modRNA therapies. Ideal carriers should have the ability to deliver modRNA to specific cell types and permit significant transgene expression for a specified duration in order to attain a satisfactory therapeutic effect, while avoiding activation from host immune responses, and/or by exhibiting ‘stealth’ features for optimal safety. A common delivery agent of choice for many RNA studies has been lipid nanoparticles (such as commercial transfection reagents RNAiMAX and MessengerMAX), which effectively express exogenous genes with high efficiency [20,26,53,54,55]. However, cytotoxic side effects of lipid nanoparticle delivery systems are common. Previous reports demonstrated that lipofectamine reagents of RNAiMAX induced a higher incidence of in vitro cell death, are also detrimental to the heart, and may cause apoptosis in vivo [20]. Furthermore, it has been reported that direct intramyocardial injections of modRNA in a clinically safe carrier containing saline/sucrose-citrate buffer can improve cardiac function in small and large animal models of heart disease (reference both the Zangi paper and the AZ paper). However, repeated injections of high doses of modRNA are associated with elevated costs and may limit clinical translations [51]. Therefore, in this study, we explored several cationic lipopolymers of PEI for modRNA delivery. The lipopolymers were previously commercialized for delivery of other nucleic acids such as siRNA [49] and pDNA [56], but they have not been explored for modRNA delivery.

### 3.1. Characterization of modRNA Complexes

We explored the capacity at which four lipopolymer carriers could safely and efficiently deliver modRNA in both in vitro and in vivo model systems. MessengerMAX was chosen as a reference carrier since its transfection efficiency was reported to be as high as ~80% for in vitro cells of human fibroblasts [26]. We first characterized the physiochemical properties of complexes formed with the GFP modRNA. As shown in Figure 1A and Appendix A, the complexes from the lipopolymers had an average hydrodynamic diameter between 150 and 240 nm, which are sufficiently small for effective cellular internalization. Furthermore, the zeta potential values of the lipopolymers reduced slightly after binding to modRNA, but they remained electropositive with charge values ranging from 25 to 35 mV, showing that these lipopolymer complexes have good stability and can penetrate the cell membrane barrier (Figure 1A). The polydispersity index (PDI) of modRNA complex particles was typically between 0.25 and 0.4, which indicated that the particles remained relatively uniform with no major aggregation and were relatively homogenous in size (Figure 1B). The size and PDI of MessengerMAX complexes were also similar to the values reported with the polymeric modRNA complexes. To further verify the conformational stability of modRNA, CD spectrum analysis was employed, where the naked modRNA and modRNA complexes shared similar conformations at ~270 nm peak (indicated by red dashed squares in Figure 1C). This demonstrated that modRNA in complexes had similar confirmation as free modRNA and should remain non-degradative and functional following release from the complexes. To assure that the polymer particles can protect the modRNA from enzymatic degradation, the modRNA complexes were incubated in 10% FBS at 37 °C for 24 h and analyzed by agarose gel electrophoresis (Figure 1D). The naked modRNA was completely degraded by the serum nucleases in FBS (indicated by white arrows in Figure 1D), while modRNA samples complexed with the polymeric carriers remained non-degradative (indicated by red dashed squares in Figure 1D).

The nature of the modRNA complexes was further analyzed with TEM and the results of analysis from a representative carrier (ALL-Fect) are shown in Figure 1E,F. The complexes were well dispersed, and the morphologies observed were mostly spherical (top right corner of Figure 1E). Irregular and elongated shapes were also found, mostly on the largest particles of the sample, which may be due to a small degree of agglomeration (indicated by white arrows in Figure 1E). Note that a certain polydispersity in complex sizes was also evident in the TEM analysis. The particles analyzed ranged from 20 to 90 nm in diameter with an average size of 38 nm (Figure 1F). The great majority of the sample (~90%) was below 60 nm and the particle population peaked at 30 nm. The average size of the ALL-Fect complexes in TEM were significantly lower than the hydrodynamic size determined by the zetasizer. The dry state of the complexes under TEM conditions and the lack of a hydration shell are the likely reasons for reduced size measurements under this analysis. This series of characterization experiments clearly indicated that lipopolymeric carriers had the right properties for delivery of modRNA to the cells.

### 3.2. In Vitro Screening for modRNA Delivery

To investigate the transfection efficiency of modRNA delivered by the chosen carriers, we initially transfected HeLa cells with a GFP modRNA at different mixing ratios (1:1 to 1:3), where different amounts of carriers were mixed with a fixed amount of modRNA (2 μg). The cells were incubated with the complexes for 24 h and analyzed by fluorescence microscopy (Figure 2A,B) and flow cytometry (Figure 2C–G). Based on the flow cytometry quantitation of mean GFP expression (Figure 2G), the transfection efficiency with polymeric carriers was comparable to that of Messenger-MAX at the optimal ratio (2:5). Particularly, ALL-Fect and Leu-Fect C had the highest delivery efficiency that reached ~83% at 3:1 ratio with these reagents. As expected, naked modRNA did not give any GFP expression in the absence of a carrier (Figure 2A). Therefore, through screening experiments, we resoundingly selected ALL-Fect and Leu-Fect C reagents for further evaluation, based on better encapsulation efficiency and delivery capacity of modRNA.

To further explore the performance of ALL-Fect and Leu-Fect C, we conducted similar experiments in 3T3 and human induced pluripotent stem cells-derived cardiomyocytes (hiPSC-CMs) in addition to HeLa cells using GFP expressing modRNAs (eGFP and nGFP), as well as a modRNA expressing mCherry. As can be seen from Figure 3, ALL-Fect and Leu-Fect C reagents could deliver several modRNA gene candidates (eGFP, nGFP, mCherry) to 3T3 and hiPSC-CM cells in addition to the HeLa cells with good transfection efficiency and high fluorescence intensity. Of interest, the GFP-transfected cardiomyocytes were still beating naturally 24 h later, suggesting that the lipopolymeric carriers could deliver mRNA to non-regenerative cells without any obvious effects on their normal physiological activity (Appendix A). In addition, these novel carriers can efficiently transfect human breast cancer MCF-7 cells, as shown in Appendix A. Thus, these results show that the selected polymeric carriers displayed potent transfection efficiency in culture with a multitude of cell types (both cell lines and primary cells), suggesting their alternative use as promising leads for therapeutic applications in a clinical setting.

### 3.3. In Vitro Biocompatibility

To verify the biocompatibility of the chosen carriers, we first tested their cytotoxicity through a series of cell culture experiments. The cytotoxic effects of the carriers were evaluated by GFP modRNA transfection in HeLa cells, using the CCK-8 assay which provides a measure of the total metabolic activity of the cells (Figure 4). Under optical microscopy (Figure 4A), the morphological characterization of cells treated with ALL-Fect and Leu-Fect C complexes showed no significant changes as compared to untransfected control groups. However, cells transfected with MessengerMAX displayed slower growth and apoptotic bodies were observed. This was consistent with the outcome of the CCK8 assay (Figure 4B), which revealed a significant decrease in proliferation rates of MessengerMAX transfected cells. On the other hand, the cells treated with ALL-Fect and Leu-Fect C complexes revealed no significant difference in proliferation capacity, as compared to the untreated cells (Figure 4B).

Next, we conducted an EdU proliferation assay to further assess the biocompatibility of the carriers. The proliferation of HeLa cells was assessed by EdU staining after 1, 2, and 3 days post-seeding and transfection (Figure 4C–E). A significant increase in the number of EdU-positive cells was seen in both ALL-Fect and Leu-Fect C complexes when compared to the cells treated with MessengerMAX complexes (Figure 4F–H). No noticeable differences of EdU incorporation were seen among the control groups and the All-Fect/Leu-Fect C complexed groups. This result further confirmed that the transfection of mRNA with the lipopolymeric carriers All-Fect and Leu-Fect C displayed low cytotoxic effects and did not impede cellular proliferation.

### 3.4. Cellular Uptake and Endosomal Escape

Following intracellular uptake, subsequent endosomal/lysosomal escape of the complexes is necessary for successful gene expression. To track intracellular uptake and localization of modRNA complexes in cells, Cy3 fluorophore was tagged to GFP modRNA so we could follow the intracellular localization of modRNA complexes. In Figure 5A, the GFP modRNA complexed with MessengerMAX (as representative analysis) revealed a significant presence of red fluorescence (Cy3-GFP modRNA complexes). These complexes were localized predominantly in the cytoplasm at 24 h as compared to 4 h, which indicated that these complexes were successfully endocytosed into the cytoplasm over time and released GFP modRNA for functional translation (given by RED GFP signals). The maximal uptake was seen 30 and 48 h after incubation of the cells with the Cy3-modRNA complexes. From Appendix A, it was also demonstrated that ALL-Fect and Leu-Fect C complexes could be endocytosed and enter the cytoplasm to translate the modRNA cargo.

The escape of the modRNA complexes from the endosomes/lysosomes was explored next. Luciferase modRNA was labeled with Cy3 and LysoTracker DND-26 (green) was used to inspect the co-localization of modRNA and lysosomal bodies (Figure 5B and Appendix A). The red (representing Luciferase modRNA) and green (representing lysosomes) fluorescent areas overlapped less frequently with increasing incubation time, which indicated that the internalized modRNA complexes successfully escape from the endosomal/lysosomal compartment. These results indicated that the polymeric complexes can convey modRNA to evade the lysosome and enter the cytoplasm to synthesize the target protein from the modRNA expression.

### 3.5. In Vivo Evaluation of modRNA Delivery

Encouraged by the suitable biocompatibility and effective transfection results in vitro, we next investigated PEI-modRNA delivery in a mouse model. Previous studies have shown that modRNA therapy prevents cardiomyocyte death and induces neovascularization with minimal side effects [21]. However, this experiment required multiple injections of large amounts of modRNA to secure adequate protein expression. To further explore the therapeutic effects of modRNA with the current carriers, modRNA encoding for the Cre reporter protein was used to evaluate the expression of modRNA by immunohistochemistry. First, primary fibroblasts derived from Cre-LoxP reporter mice were used to assess the expression of modRNA coding for tdTomato (Figure 6A). Efficient tdTomato expression was obtained by the modRNA delivered by all 3 vectors; MessengerMAX, ALL-Fect, and Leu-Fect C (Figure 6B,C). Next, we used insulin syringes to inject a low dose of modRNA complexes directly into the myocardium of Cre-LoxP reporter mice, where the tdTomato gene expression is activated upon excision of LoxP locus by cyclization recombinase (Figure 6A). All 3 carriers efficiently delivered functionally active modRNA and expressed the target protein (given by red tdTomato expression) after 3 days of local injection (Figure 6D). Based on analysis of the tdTomato-positive cell population (Figure 6E), Leu-Fect C delivering modRNA was superior to MessengerMAX and ALL-Fect carriers.

Intravenous (IV) injection is considered to be an efficient way to induce target protein expression in the liver and spleen, since most nano-sized complexes are typically deposited into the liver and spleen from the systemic circulation [4]. To test this approach, we administered a low dose of Cre modRNA/polymer complexes (20 μg of Cre modRNA) through the tail vein of Cre-LoxP reporter mice and assessed the efficiency of transfection in spleen and livers after 3 days (see schematic, Figure 7A,D). Using the same methodology as above, a strong tdTomato fluorescence was detected in target organ tissues of spleen (Figure 7B) and liver (Figure 7E). Interestingly, the delivery efficiency by the Leu-Fect C complexes was superior to both ALL-Fect polymer and MessengerMAX control, based on the expression pattern resulting from Cre activity in both the spleen (Figure 7C) and liver (Figure 7F), respectively. Of interest, very limited or no tdTomato expression was observed in heart and kidney following tail vein injections (Appendix A). These results imply potential liver and splenic tissue tropism of Leu-Fect C mRNA delivery following systemic administration. Furthermore, these results indicated that the lipopolymer carriers maintain strong encapsulation properties and protect modRNA during systemic circulation. Ultimately, the desired in vivo-produced protein can be effectively expressed with a relatively small dose of administered modRNA.

### 3.6. In Vivo Biocompatibility

To evaluate the safety of polymer carriers in vivo, we first tested the local tolerability of the carriers by evaluating the effects of local injection (50 μL) into the pancreas, a major secretory organ of the body (Figure 8A). At three days post-administration, the pancreas was harvested and processed for H&E staining (Figure 8B). Examination of the tissue sections revealed that pancreatic tissues receiving MessengerMAX displayed severe tissue necrosis, while the pancreases receiving lipopolymeric carriers exhibited normal tissue morphologies without any signs of necrosis or apoptosis. These results add further credence to the excellent biosafety of the ALL-Fect and Leu-Fect C reagents in vivo. Next, we stained the sections to assess for the presence of infiltrating CD3^+^ and CD68^+^ lymphocytes and monocytes, indicators of local immune responses (Figure 8C,D). Indeed, quantitative analysis revealed a significant increase in the number of CD3^+^ and CD68^+^ cells in the pancreases receiving MessengerMAX injection, while polymer injections to the pancreas with ALL-Fect and Leu-Fect C showed no discernable differences to the control (Figure 8E,F).

Next, we further explored the biosafety of the delivery systems by repeat administration into the tail vein once every 2 days over the course of 2 weeks, for a total of 7 times (Figure 9A). Body weights of mice were analyzed as an indicator of general treatment-induced toxicity as a study endpoint. As seen in Figure 9B, no significant body weight loss was observed in any group during the treatments, indicating unnoticeable adverse effects in this delivery mode. Systemic toxicity was also assessed by histopathology and blood serum chemistry. Based on the HE staining results (Figure 9C), there were no pronounced histological abnormalities found in the heart, liver, spleen, lung, kidney, or intestine tissues of mice following administration of the different administered carrier groups. Based on the analysis of cells associated with inflammation, the population of WBC (white blood count) cells and lymphocytes (Lymph) was significantly increased in the MessengerMAX treatment group compared to all other groups (Figure 9D,E). Repeated administration with the MessengerMAX treatment group also elevated levels of monocytes (Mon population) compared to no treatment/baseline levels (Figure 9F), but this elevation was not significantly different from the other treatment groups. No clear distinction between the granulocyte (Gran) cells was evident among the study groups, as shown in Figure 9G. The hematological analysis also showed normal liver function with levels of AST and ALT in the normal range for all carriers (Figure 9H,I). Based on this analysis, the novel lipopolymeric vectors displayed significant advantages over the commercial liposomal reagent MessengerMAX in terms of their tolerability in vivo, which confirm a satisfactory safety profile for further animal testing en route to the clinical application of modRNA therapeutics.

## 4. Conclusions

Our in vitro study results highlighted excellent transfection efficiencies when transfecting modRNA in cells using ALL-Fect and Leu-Fect C polymer carriers. Transfection efficiencies reached >80% at the optimal ratio of 3:1, yielding higher delivery efficiency as compared to the other polymeric carriers. Further investigation showed that these two novel reagents can mediate endosomal escape and release of modRNA cargo, which may be due to the small particle size of the resultant complexes. Importantly, ALL-Fect and Leu-Fect C demonstrated potent delivery capability in vivo at low dose administration. We also observed ideal biocompatibility and tolerability in mouse models with no adverse effects under the study conditions. The toxicity studies and histopathology analysis revealed that repeated administration is well tolerated even at higher doses, where no evidence of tissue necrosis or organ failure was documented. Of interest were our suggestive findings that alluded to lipopolymer tissue tropism in spleen and liver. More experiments are needed in order to decipher the biological mechanisms underpinning this translocation. Furthermore, our findings may suggest that these lipopolymer carriers could be more potent for modRNA delivery compared to native polymers or other commercial transfection reagents. This is the first report on the use of these new lipopolymer carriers to effectively deliver modRNA in animal models and these carriers are expected to play an important role in the future delivery of modRNA for gene therapy. Therefore, this study may open up new avenues for modRNA delivery in vivo and provide great hope to generate a new avenue for the targeted treatment of human diseases.

## Figures and Tables

**Figure 1 pharmaceutics-15-00410-f001:**
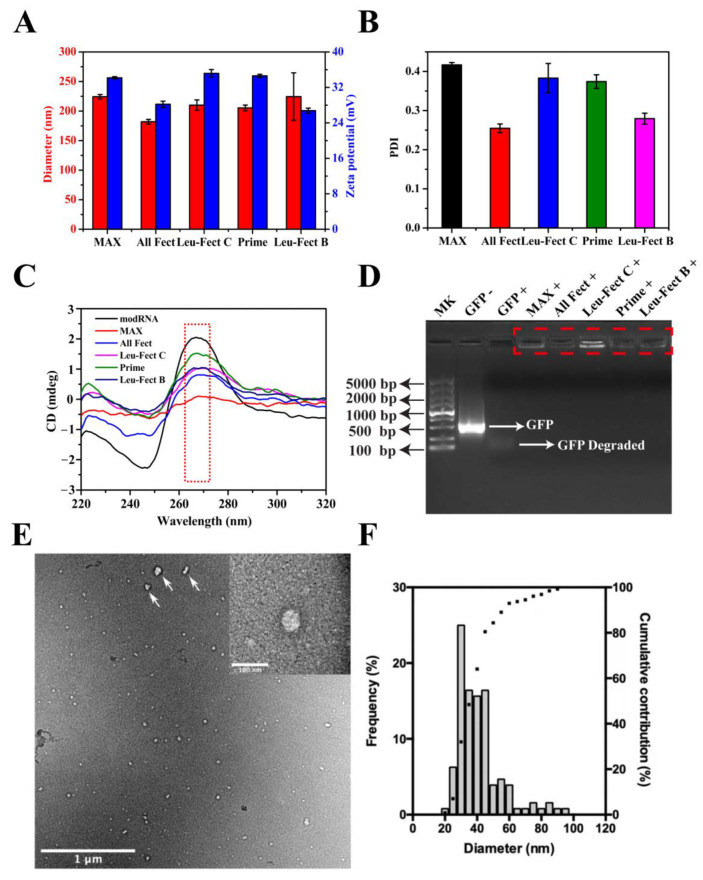
Characterization of complexes obtained from lipopolymers. (**A**) Size distribution and zeta potential of modRNA complexes with indicated polymers measured by the dynamic light scattering. Data represent mean ± SD (*n* = 3). (**B**) The PDI of modRNA/polymer complexes. Data represent mean ± SD (*n* = 3). (**C**) Conformation stability of modRNA complexes with various polymer by circular dichroism spectrometry. (**D**) Agarose gel electrophoresis assay of modRNA stability in vectors.+: contains 10% serum, -: no serum. (**E**) TEM micrograph of ALL-Fect complexes formed with modRNA (modRNA: ALL-Fect ratio of 1:5; scale bars = 1 µm and 100 nm in insert) and (**F**) particle size distribution obtained from representative TEM micrographs.

**Figure 2 pharmaceutics-15-00410-f002:**
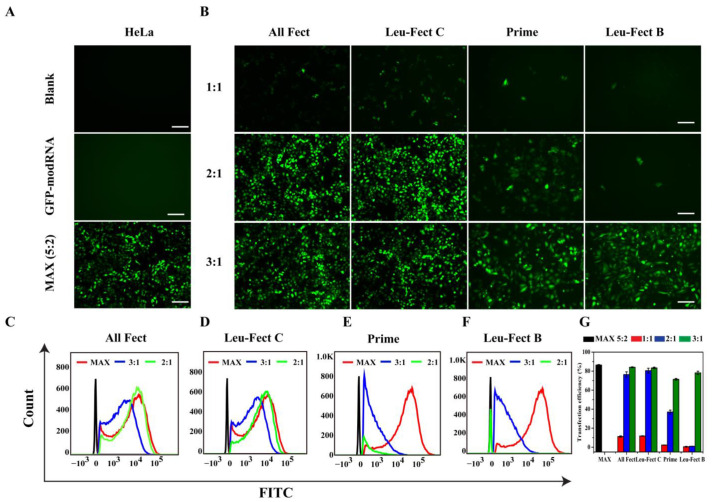
Screening of polymeric vectors in HeLa cells. (**A**) GFP modRNA transfection by MessengerMAX reagent at 5:2 ratio and naked modRNA as a control group. (**B**) GFP modRNA transfection by the polymeric vectors at different ratios. The scale bars in (**A**,**B**) represent 200 μm. (**C**–**F**) The transfection efficiency was analyzed by flow cytometry. Red represents the transfection efficiency of MessengerMAX at the optimal ratio of 5:2, blue represents the transfection efficiency of the polymer at a ratio of 3:1, and green the transfection efficiency of the polymer at a ratio of 2:1. (**G**) Transfection efficiency statistics for different vectors and modRNA at different mixing ratios. Data represent mean ± SD (*n* = 3).

**Figure 3 pharmaceutics-15-00410-f003:**
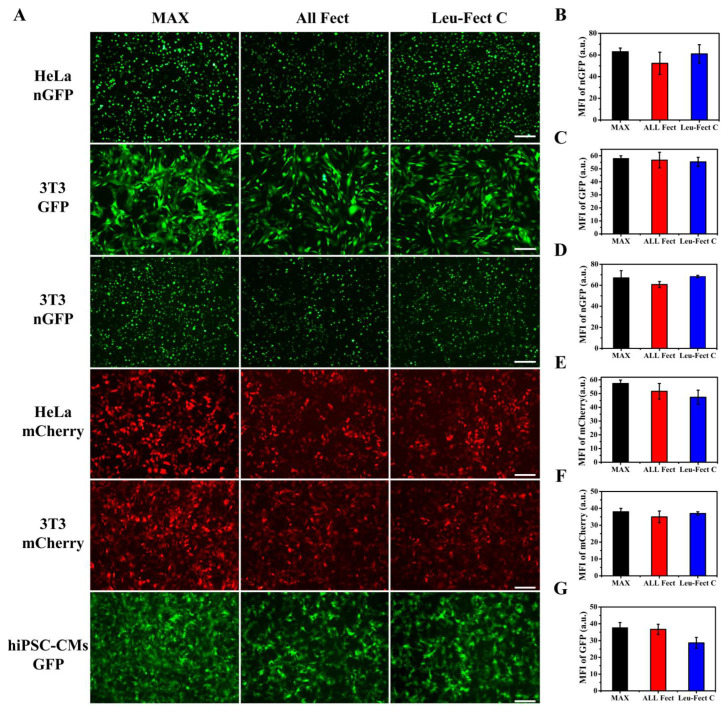
Assessment of lipopolymer modRNA delivery to mammalian cell types. (**A**) The transfection expression of different modRNA at the optimal ratio was evaluated in HeLa, 3T3, and hiPSC-CMs. The scale bar represents 200 μm. (**B**–**G**) The mean fluorescence intensity (from Image J software version 1.8.0 analysis) for different cells after delivery of modRNA by the indicated vectors.

**Figure 4 pharmaceutics-15-00410-f004:**
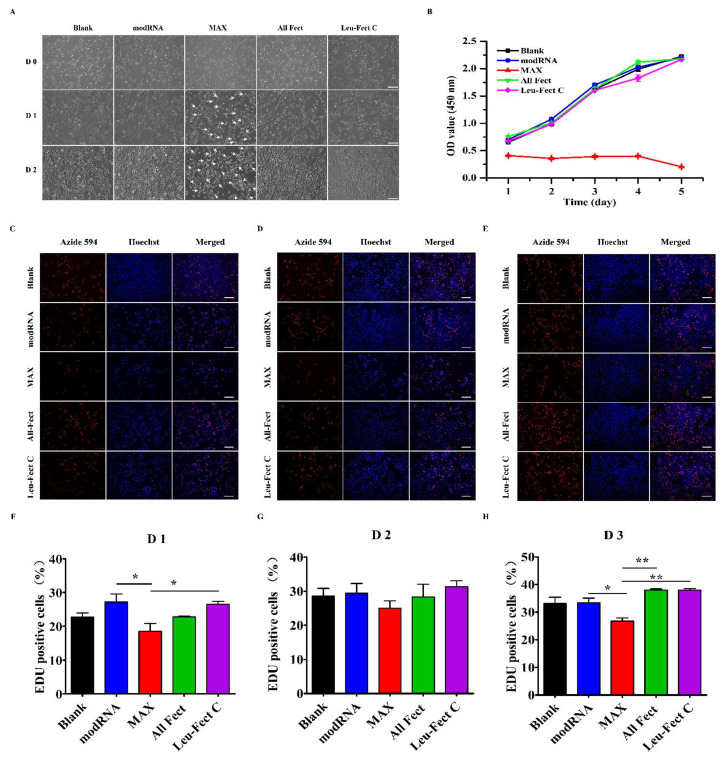
In vitro safety of lipopolymer vectors. (**A**) The morphology of cells after treatment with chosen vectors (scale bar: 200 μm). (**B**) Cellular proliferation after treatment with lipopolymer complexes at daily time intervals, as determined by the CCK-8 assay. Data represent mean ± SD (*n* = 5). (**C**–**E**) EdU stained cells at different time points. EdU (red) and Hoechst (blue) show the nuclei of cells in the cell cycle and all nuclei, respectively (scale bar: 200 μm). (**F**–**H**) Quantitative analysis of EdU-positive cells at 1, 2 and 3 days. (*n* = 5) (* *p* < 0.05, ** *p* < 0.01).

**Figure 5 pharmaceutics-15-00410-f005:**
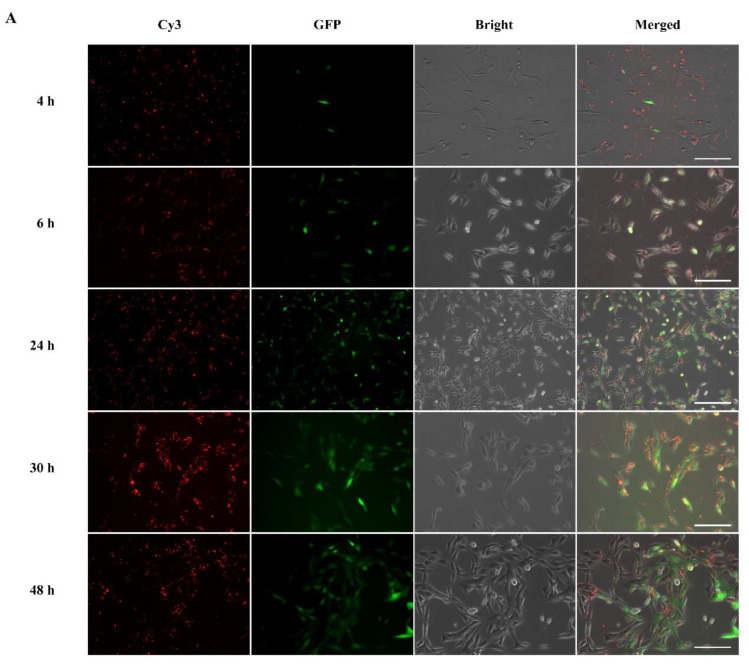
Cellular uptake route and intracellular localization of modRNA polymer complexes. (**A**) The GFP modRNA complex enters the cytoplasm over time, where it translates and expresses the target protein in 3T3 cells. The scale bar represents 200 μm. (**B**) Intracellular uptake images of Luciferase-modRNA polymer complexes in HeLa cells after 2 h, where cell nuclei were stained blue, lysosome was stained green, and modRNA was stained red. Scale bar is 25 µm.

**Figure 6 pharmaceutics-15-00410-f006:**
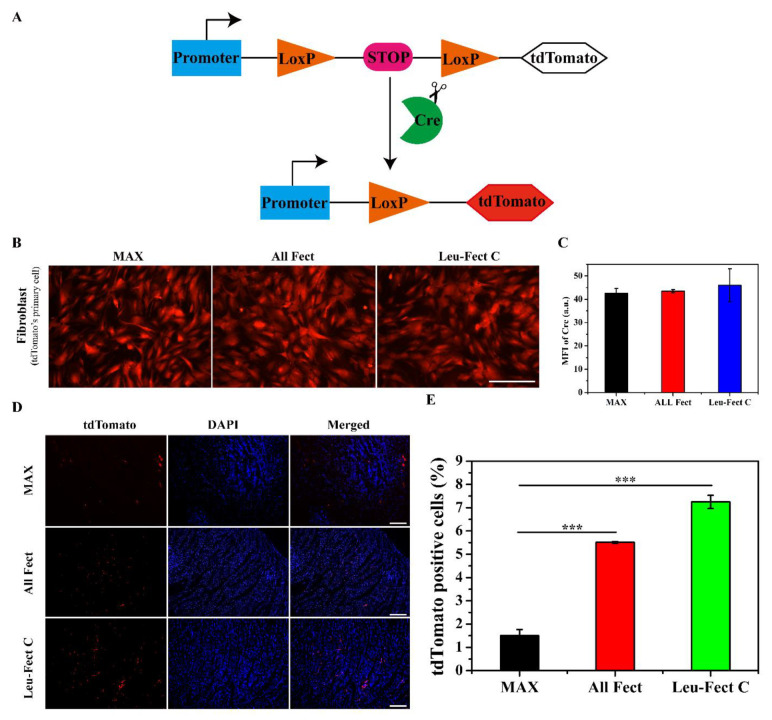
In vivo investigation of local organ injection of modRNA. (**A**) Cre/LoxP system working principle. (**B**) Fluorescent pictures of tdTomato expression following the transfection of Cre modRNA in primary mouse fibroblast cells (scale bar: 200 µm). (**C**) The mean fluorescence intensity of Cre modRNA expression of tdTomato in cells. (**D**) The fluorescent images of tdTomato expression levels in mouse hearts after 3 days of modRNA administration (scale bar: 200 µm). (**E**) Quantitative analysis of tdTomato-positive cell population in hearts. (*** *p* < 0.001).

**Figure 7 pharmaceutics-15-00410-f007:**
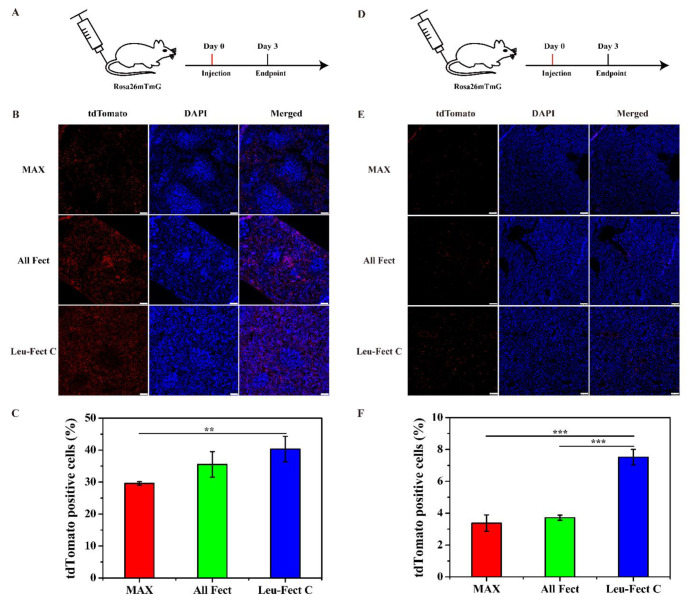
In vivo evaluation of modRNA delivery by tail vein injection. (**A**,**D**) Schematic depiction of in vivo transfection of Cre modRNA delivered by tail vein injection. (**B**,**E**) Three days after modRNA polymer complexes were injected into the tail vein; frozen sections of the spleen and liver were taken for staining to observe the effect of modRNA expression. Immunofluorescence picture of in vivo transfection efficiency of Cre-modRNA in the spleen and liver. Scale bar: 50 μm. (**C**,**F**) The statistical chart of tdTomato positive cells expression in spleen and liver. (Total *n* = 3 mice, ** *p* < 0.01, *** *p* < 0.001).

**Figure 8 pharmaceutics-15-00410-f008:**
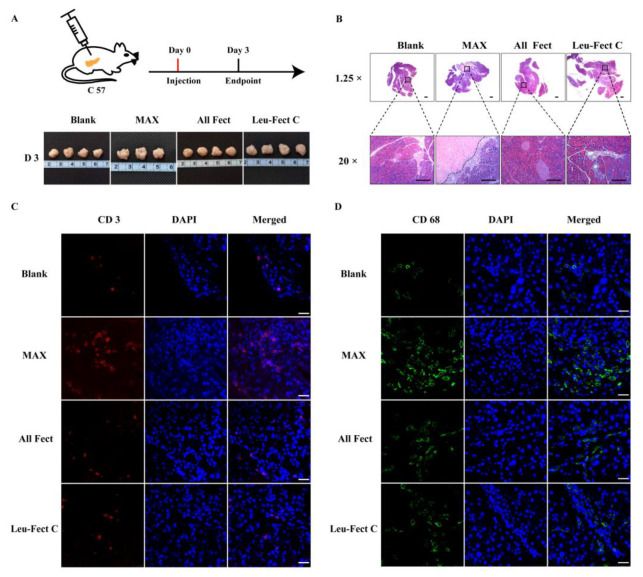
In vivo biocompatibility of the polymeric vectors in the pancreatic organ. (**A**) A mixture of 50 μL vector (1 μg/μL) and 50 μL serum-free Opti-MEM medium was evenly injected into the pancreas and the organs were harvested after 3 days (*n* ≥ 3 mice). The lower part of (**A**) is the gross appearance of the pancreatic organs. (**B**) Histopathological analysis of pancreatic organs. The typical tissue sections were indicated at low (upper part) and high (lower part) magnification pictures. (The scale bar is 500 and 100 μm, respectively). (**C**,**E**) Sections were stained with CD3 and CD68 to detect the specific immune cells in the pancreatic organs. (**D**,**F**) Quantitative analysis of CD3^+^ and CD68^+^ cells in the tissue sections, respectively. (Scale bar is 25 μm for (**C**,**D**). * *p* < 0.05, *** *p* < 0.001).

**Figure 9 pharmaceutics-15-00410-f009:**
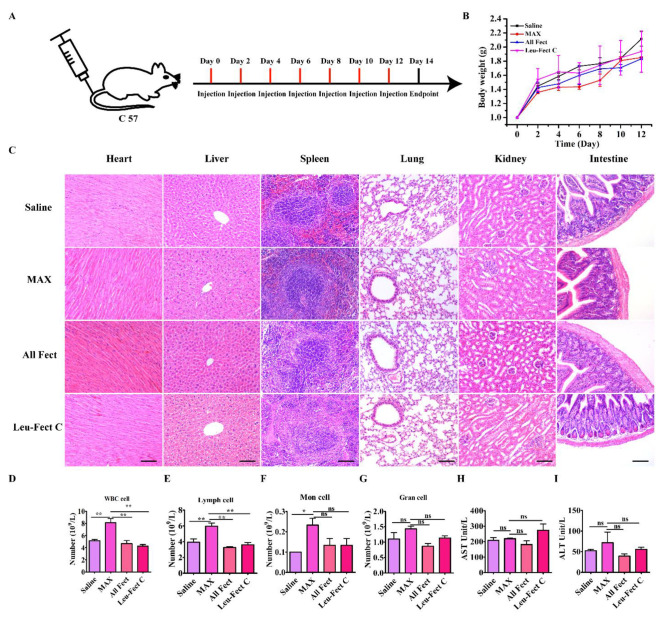
Safety profile analysis of polymeric carriers in vivo. (**A**) Schematic depicting frequency of tail vein injections (*n* = 5). (**B**) Growth curve of mouse weight after different days of treatment (*n* = 5). (**C**) Histopathological analysis of different organs after systemic delivery of diverse vectors. The scale bar is 100 µm. (**D**–**I**) Polymeric carrier biosafety profile as assessed 14 days post IV administration using biochemistry analysis of blood samples. (* *p* < 0.05, ** *p* < 0.01, ns = not significant).

## Data Availability

Data is contained within the article and supplementary material.

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
