# Peer review of "Safe and Effective Delivery of mRNA Using Modified PEI-Based Lipopolymers"

_pharmaceutics, 2023, doi:10.3390/pharmaceutics15020410_

Round 1
Reviewer 1 Report
The manuscript deals with Modified PEI Based Lipopolymer for effective delivery of mRNA. The topic is interesting; however, some concerns have to be addressed prior to publication
1. English language check throughout the manuscript is required.
2. references update is mandatory
3. Methods: The methodology of preparing the proposed delivery vector should be described in details prior to going through characterization.
4. The difference between the size obtained by zeta Sizer and TEM is quite high. Although the authors presented an explanation, this explanation can be accepted only if there is a small non-significant difference. the authors should check this again.
5. PDI low value indicated homogeneity of dispersion which could reflect to some extent its physical stability against aggregation. The authors statement about concluding the stability of the delivery system is misleading and should be clarified.
Reviewer 2 Report
This article did a lot of works and made a detailed evaluation about the PEI based Lipopolymers. This was a very comprehensive manuscript. But there still have few questions. The questions were list as below.
1: In the figure 1E, are the other small point in the TEM picture also nanoparticles? What is the other point?
2:In the figure 7E,F, Could you please explain why the tdTomato positive cells in All Fect group were lower than other two groups?
3: About the Figure 5B, Could you also please added another time points? Such as 2h, 4h, 8h and then 24h.
In order to see clearer, please also provide high zoom view with the endosomal details
Please added Pearson’s correlation coefficient.
4: In the figure 5A, please change the picture in the 48h and 30h time point with the same bar.
Reviewer 3 Report
The authors present their research on the delivery of chemically modified RNA using commercially available mRNA transfection reagents. The study was correctly designed. The spectrum of different types of analyses, both physicochemical and biological, was vast and thus the presented research is valuable. I recommend the authors should read the text of the manuscript carefully since minor spelling issues need to be addressed before publication, e.g. PDI is a polydispersity index not polymer dispersity index, CO2 is not CO2, etc.
Moreover, the authors claim that: "The polydispersity index (PDI) of modRNA complexes particles was typically between 0.25 and 0.4, which indicated that the particles remained stable Figure 1 B." Could you please explain on what basis? Usually, the stability of the particles is based on the studies performed in the predetermined set of time when the particles are suspended in the medium and the hydrodynamic diameters are measured in different time laps. Alternatively, zeta potential values for polyplexes should be measured. Can authors provide an exemplary DLS histogram for one of the polyplex samples? Were sizes measured by Intensity or Volume?
In Conclusion, the authors state that: "This the first report of the use of new polymeric carriers to effectively deliver modRNA in animal models." Could you explain how commercially available Lipofectamine™ MessengerMAX mRNA Transfection Reagent is new?
Reviewer 4 Report
The manuscript is interesting however, there are aspects that should be better explored and explained before acceptance for publication.
comments
1- The abbreviation should be defined the first time they appear and should be added in parentheses after the written-out form.
2- The abstract should be improved.
3- In the methodology part, 2.3.1 Measurement of Particle Size
Authors should write the details of the experiment like the dilution of samples before measurement, the angle of measurement, and the temperature
4- At which accelerating voltage the TEM measurement was done?
5- Authors should put references for each experiment.
6- The discussion should be compared with the previous finding related to the study.
Round 2
Reviewer 1 Report
Can be accepted in current form
Reviewer 4 Report
Manuscript was revised point by point according to reviewer comments
It is more acceptable now.